

# Evaluating Lossy Data Compression on Climate Simulation Data within a Large Ensemble

Allison H. Baker[1], Dorit M. Hammerling[1], Sheri A. Mickelson[1], Haiying Xu[1], Martin B. Stolpe[2], Phillipe Naveau[3], Ben Sanderson[1], Imme Ebert-Uphoff[4], Savini Samarasinghe[4], Francesco De Simone[5], Francesco Carbone[5], Christian N. Gencarelli[5], John M. Dennis[1], Jennifer E. Kay[6], and Peter Lindstrom[7]

[1]The National Center for Atmospheric Research, Boulder, CO, USA
[2]Institute for Atmospheric and Climate Science, ETH Zurich, Zurich, Switzerland
[3]Laboratoire des Sciences du Climat et l'Environnement, France
[4]Department of Electrical and Computer Engineering, Colorado State University, Fort Collins, CO, USA
[5]CNR-Institute of Atmospheric Pollution Research, Division of Rende, UNICAL-Polifunzionale, Rende, Italy
[6]Department of Oceanic and Atmospheric Sciences, University of Colorado, Boulder, CO, USA
[7]Center for Applied Scientific Computing, Lawrence Livermore National Laboratory, Livermore, CA, USA

*Correspondence to:* Allison H. Baker (abaker@ucar.edu)

**Abstract.** High-resolution earth system model simulations generate enormous data volumes, and retaining the data from these simulations often strains institutional storage resources. Further, these exceedingly large storage requirements negatively impact science objectives by forcing reductions in data output frequency, simulation length, or ensemble size, for example. To lessen data volumes from the Community Earth System Model (CESM), we advocate the use of lossy data compression techniques. While lossy data compression does not exactly preserve the original data (as lossless compression does), lossy techniques have an advantage in terms of smaller storage requirements. To preserve the integrity of the scientific simulation data, the effects of lossy data compression on the original data should, at a minimum, not be statistically distinguishable from the natural variability of the climate system, and previous preliminary work with data from CESM has shown this goal to be attainable. However, to ultimately convince climate scientists that it is acceptable to use lossy data compression, we provide climate scientists with access to publicly available climate data that has undergone lossy data compression. In particular, we report on the results of a lossy data compression experiment with output from the CESM Large Ensemble (CESM-LE) Community Project, in which we challenge climate scientists to examine features of the data relevant to their interests, and attempt to identify which of the ensemble members have been compressed and reconstructed. We find that while detecting distinguishing features is certainly possible, the compression effects noticeable in these features are often unimportant or disappear in post-processing analyses. In addition, we perform several analyses that directly compare the original data to the reconstructed data to investigate the preservation, or lack thereof, of specific features critical to climate science. Overall, we conclude that applying lossy data compression to climate simulation data is both advantageous in terms of data reduction and generally acceptable in terms of effects on scientific results.





# 1 Introduction

Earth system models are widely-used to study and understand past, present, and future climate states. The tremendous advances in computational power (i.e., processor speeds) over the last 25 years have allowed earth system modelers to use finer temporal and spatial model resolutions. While finer resolutions typically produce more accurate and realistic simulations, the resulting

data sets are often massive and may severely strain data storage resources. Because supercomputing storage capacities have not increased as rapidly as processor speeds over the last 25 years, the cost of storing huge data volumes is becoming increasingly burdensome and consuming larger and unsustainable percentages of computing center budgets (e.g., Kunkel et al. (2014)).

The Community Earth System Model (CESM) is a popular and fully-coupled climate simulation code (Hurrell et al., 2013) whose development is led by the National Center for Atmospheric Research (NCAR). The CESM regularly produces large data

sets resulting from high-resolution runs and/or long timescales that strain NCAR storage resources. For example, to participate in the Coupled Model Comparison Project Phase 5 (CMIP5, 2013) that led to the Intergovernmental Panel on Climate Change (IPCC, 2016) Assessment Report 5 (AR5) (Stocker et al., 2013), CESM produced nearly 2.5 PB of raw output data that were post-processed to obtain the 170 TB of data submitted to CMIP5 (Paul et al., 2015). Current estimates of the raw data requirements for CESM for the up-coming CMIP6 project (Meehl et al., 2014) are in excess of 10 PB (Paul et al., 2015).

A second example of a data-intensive CESM project is the CESM-Large Ensemble (LE) Project (Kay et al., 2015), a large ensemble climate simulation study. The CESM-LE Project is a publicly available collection of 180-year climate simulations at approximately 1-degree horizontal resolution for studying internal climate variability. Storage constraints influenced the frequency of data output and necessitated the deletion of the raw monthly output files. In particular, the initial 30 ensemble member simulations generated over 300 TB of raw data, and less than 200 TB of processed and raw data combined could

be retained due to disk storage constraints. For large climate modeling projects such as CMIP and CESM-LE, reducing data volumes via data compression would mitigate the data volume challenges by enabling more (or longer) simulations to be retained, and hence allow for more comprehensive scientific investigations.

The impact of data compression on climate simulation data was addressed in Baker et al. (2014). In Baker et al. (2014), quality metrics were proposed to evaluate whether errors in the reconstructed CESM data (data that had undergone compression)

were smaller than the natural variability in the data induced by the climate model system. The results of the preliminary study indicated that a compression rate of 5:1 was possible without statistically significant changes to the simulation data. While encouraging, our ultimate goal is to demonstrate that the effect of compression on the climate simulation can be viewed similarly to the effect of a small perturbation in initial conditions or running the exact same simulation on a different machine. While such minor modifications lead to data that are not bit-for-bit (BFB) identical, such modifications should not result in an altered

climate (Baker et al., 2015). With compression in particular, we must also ensure that nothing systematic (i.e., over-smoothing) has been introduced. Therefore, to build confidence in data compression techniques and promote acceptance in the climate community, our aim in this work is to investigate whether applying lossy compression impacts science results or conclusions from a large and publicly available CESM dataset.





To this end, we provided climate scientists with access to climate data via the CESM-LE project (Kay et al., 2015). We contributed three additional ensemble members to the CESM-LE project and compressed and reconstructed an unspecified subset of the additional three members. To determine whether the effects of compression could be detected in the CESM-LE data, we then enlisted several scientists to attempt to identify which of the new members had undergone lossy compression by

using an analysis technique of their choosing (i.e., we did not specify what analysis technique each should use). In addition, we provided a different group of scientists with both the original and reconstructed datasets and asked them to directly compare features particular to their interests (again, we did not specify how this analysis should be done) and determine whether the effects of compressing and reconstructing the data impacted climate features of interest. Indeed, a significant contribution of our work was enabling scientists to evaluate the effects of compression on any features of the data themselves with their

own analysis tools (rather than relying solely on simple error metrics typically used in compression studies). Note that while the three additional CESM-LE ensemble members were generated at NCAR, the scientists participating in the ensemble data evaluations were from both NCAR and external institutions. The author list for this paper reflects both those who conducted the study as well as those who participated in the lossy data evaluations (and whose work is detailed in this paper). For simplicity, the term "we" in this paper can indicate any subset of the author list, and in Appendix A we detail which authors conducted

each of the data evaluations described in this work.

In this paper, we describe several of the analyses done by scientists and detail the results and the lessons that we learned from their investigations. We demonstrate the potential of lossy compression methods to effectively reduce storage requirements with little to no *relevant* information loss, and our work sheds light on what remains to be done to promote widespread acceptance and use of lossy compression in earth system modeling. This paper is organized as follows. We first discuss background

information in Sect. 2. In Sec. 3, we describe our approach to demonstrating the effects of lossy compression on climate science results. Then, in Sect. 4 and 5, we present selected results from data analyses evaluating compression effects in the context on the CESM-LE data. Finally, we summarize the lessons learned from this study in Sect. 6 and offer concluding thoughts in Sect. 7.

## 2 Background

In this section, we further discuss lossy data compression. We then provide additional details on the CESM-LE project datasets.

### 2.1 Data compression

Compression techniques are classified as either lossless or lossy. Consider a dataset $\mathcal{X}$ that undergoes compression, resulting in the compressed dataset $\mathcal{C}$ ( $\mathcal{X} \Rightarrow \mathcal{C}$). When the data is *reconstructed*, then $\mathcal{C} \Rightarrow \tilde{\mathcal{X}}$. If the compression technique is lossless, then the original data is exactly preserved: $\mathcal{X} = \tilde{\mathcal{X}}$. Note that the commonly-used *gzip* compression utility is a lossless method.

If, on the other hand, the compression technique is lossy, then $\mathcal{X} \approx \tilde{\mathcal{X}}$; the data is not exactly the same (e.g., Sayood (2012)). Lossy compression methods generally give the user some control over the information loss via parameters that either control the compression rate, precision, or absolute or relative error bounds. The effectiveness of compression is generally measured





by a compression ratio (CR), which is the ratio of the size of the compressed file to that of the original file (c.f. Iverson et al. (2012)):

$$\mathrm{CR}(F) \;=\; \frac{\mathrm{filesize}(\mathcal{C})}{\mathrm{filesize}(\mathcal{X})}. \tag{1}$$

While lossless methods are often viewed as "safer" for scientific data, it is well known that lossless data compression
of floating point simulation data is difficult and often yields little benefit (e.g., Lindstrom and Isenburg (2006), Bicer et al. (2013), Lakshminarasimhan et al. (2011)). The reason for the relative ineffectiveness of lossless methods on scientific data (in contrast to image or audio data, for example) is that trailing digits of the fixed precision floating-point output data are often essentially random, depending on the data type and the number of physically significant digits. Random numbers are a liability for compression, thus giving lossy methods a significant advantage. Many recent efforts have focused on effectively
applying or adapting lossy techniques for scientific datasets (e.g., Lakshminarasimhan et al. (2011), Iverson et al. (2012), Bicer et al. (2013), Laney et al. (2013), Gomez and Cappello (2013), Lindstrom (2014)). However, a major obstacle inhibiting the adoption of lossy compression by many scientific communities is not technical, but rather psychological in nature. In the climate modeling community, scientists who analyze the simulation data are often (understandably) reluctant to lose bits of data in order to achieve smaller data volumes. In remarkable contrast, meteorological communities widely use and trust the World
Meteorological Organization (WMO) accepted GRIB2 (Day et al., 2007) file format, which encodes data in a lossy manner. It should be noted, however, that difficulties can arise from GRIB2's lossy encoding process, particularly with new variables with large dynamic ranges or until official GRIB2 specification tables are released for new model output (see, e.g., GFAS, 2015). While the preliminary work in Baker et al. (2014) indicated that GRIB2 was not as effective as other compression methods on CESM data, a more extensive investigation of GRIB2 with climate data should be done in light of the new techniques
in Baker et al. (2015) and this paper before definitive conclusions are drawn. Nevertheless, the contrast is notable between the meterological community's wide-spread use and acceptance of GRIB2 and the climate community's apparent reluctance to adopt lossy methods, even when proven to be safe, flexible and more effective. In this context, when applying lossy compression to scientific datasets, determining appropriate levels of precision or error, which result in only a negligible loss of information, is critical to acceptance.

In summary, there are several salient points to recognize in the case for adopting lossy compression for climate simulation data. First, the least few significant bits of data are usually noise resulting from the fixed-precision rounding error and are not physically meaningful. Second, while 32-bit and 64-bit are meaningful data sizes for hardware, those sizes have no inherent relevance to a particular climate simulation. In other words, there is not a compelling reason why 32-bits is the most accurate representation for a particular variable on a particular grid resolution (e.g., consider saving fewer bits from a finer resolution
versus saving more bits from a coarser resolution). Finally, note that regardless of the precision of the simulation output data, this data has already been subjected to a lossy process via the chosen output frequency (e.g., hourly, daily, monthly). Therefore, we argue that applying lossy compression to climate simulation data should not be regarded with more suspicion than carefully choosing grid resolutions, output frequency, and computation precisions.



## 2.2 The CESM Large Ensemble project dataset

The CESM-LE project (Kay et al., 2015) is a community project that includes a publicly available ensemble of climate model simulations generated for the purpose of studying internal climate variability. All data are currently available from the Earth System Grid website (http://www.earthsystemgrid.org). The CESM-LE project is an ideal venue for this evaluation because of

its use of climate ensembles, struggle with storage limitations, and availability to the broader climate community. The project began with a set of thirty ensemble members, each of which covers the period from 1920-2100. All simulations use the fully-coupled 1-degree latitude/longitude version of CESM-CAM5. Historical forcing is used for the period 1920-2005 and RCP8.5 radiative forcing (i.e., forcing that reflects near past and future climate change, e.g. Lamarque et al. (2011)) thereafter. Ensemble spread is generated using small round-off level differences in the initial atmospheric temperature field. Comprehensive details

on the experimental setup can be found in Kay et al. (2015).

CESM outputs raw data in NetCDF-formatted time-slice files, referred to as "history" files, for post-processing analysis. Sample rates (e.g. daily, monthly, etc.) are determined for each variable by default, depending on the grid resolution, though a user can specify a custom frequency if desired. When the floating-point data are written to these history files, they are truncated from double-precision (64-bits) to single-precision (32-bits). For the CESM-LE project, monthly, daily, and 6-hourly history

file outputs were converted and saved as single variable timeseries, requiring approximately 1.2 TB of storage per ensemble member. Complete output variable lists and sampling frequencies for each model can be found at https://www2.cesm.ucar.edu/models/experiments/LENS/data-sets. We restrict our attention in this work to data from the atmospheric model component of CESM, which is the Community Atmosphere Model (CAM). CAM output data for the CESM-LE simulations consists of 159 distinct variables, many of which are output at multiple frequencies: 136 have monthly output, 51 have daily output, and 25

have 6-hourly output (212 total variable outputs). Note that due to storage constraints, the 6-hourly data are only available during three time periods: 1990-2005, 2026-2035, 2071-2080.

## 3  Approach

To provide climate scientists with the opportunity to determine whether the effects of lossy compression are detectable and to solicit community feedback, we first designed a blind evaluation study in the context of the CESM-LE project. Three new

simulation runs were setup identically to the original 30, differing only in the unique perturbation to the initial atmospheric temperature field. We then contributed these three new additional ensemble members (labeled 31-33) to the CESM-LE project, first compressing and reconstructing the atmospheric data output from two of the new ensemble runs (31 and 33). By not specifying which of the new ensemble members (or how many) had been subject to compression, we were able to gather feedback from scientists in the climate community detailing which ensemble member(s) they believed to have been compressed

and why. We did not specify how the data should be analyzed. In the context of the CESM-LE project, we were able to question whether the effects of compression could be distinguished from model internal variability. In addition, we supplied several scientists with both the original and reconstructed data for ensemble members 31 and 33, allowing direct comparison of the two.



Based on its superior performance on the climate data in Baker et al. (2014), we chose the publicly available *fpzip* algorithm (Lindstrom and Isenburg, 2006) for lossy data compression. The *fpzip* algorithm is particularly attractive because it is fast at both compression and reconstruction, freely available, grid independent, and can be applied in both lossless and lossy mode. The *fpzip* method uses predictive coding, and its lossy mode is invoked by discarding a specified number of least significant

bits before losslessly encoding the result, which results in a bounded relative error.

The diverse nature of climate model data necessitates determining the appropriate amount of compression (i.e., parameter) on a per-variable basis (Baker et al., 2014). Some variables can be compressed more aggressively than others, and the appropriate amount of compression can be influenced by characteristics of the variable field and properties of the compression algorithm. For example, relatively smooth fields are typically easy to compress, whereas fields with jumps or large dynamic ranges often

prove more challenging. Further, if the internal variability is large for a particular variable across the ensemble, then more compression error can be tolerated. With *fpzip*, controlling the amount of compression translates to specifying the number of bits of precision to retain for each variable timeseries. Note that if a variable is output at more than one temporal frequency, we do not assume that the same precision will be used across all output frequencies. Recall that the CAM timeseries data in CESM-LE contain single-precision (32-bit) output. While one could specify that *fpzip* retains any number of bits (up to 32),

we restrict our choices to 16, 20, 24, 28, and 32, the latter of which is lossless for single-precision data.

In Baker et al. (2014), the appropriate level of compression was chosen for each of the CAM variables in the dataset by selecting the most aggressive (lowest CR) such that a suite of four quality metrics all passed. The quality metrics in Baker et al. (2014) are largely based on evaluating the error in the reconstructed dataset in the context of an ensemble of simulations and test for variables for Z-score, maximum pointwise error, bias, and correlation. The ensemble distribution is intended to represent

acceptable internal variability in the model, and the goal is that the error due to lossy compression should not be distinguishable from the model variability as represented by the ensemble distribution. Note that for some variables, the lossless variant of a compression algorithm was required to pass the suite of metrics. (In the case of *fpzip*, the lossless variant was required for less than five percent of the variables.) While many of the variables present in the CAM dataset in Baker et al. (2014) are also present in the CESM-LE dataset studied here, we did not necessarily use the same *fpzip* parameter settings for the variables

common to both for several reasons. First, the data in Baker et al. (2014) were output as annual averages, which we would expect to be smoother (and easier to compress) than the 6-hourly, daily and monthly data from CESM-LE. Also, the choices of bits to retain with *fpzip* in Baker et al. (2014) were limited to 16, 24, and 32. And notably, the CAM variant in Baker et al. (2014) used the spectral element (SE) dynamical core, whereas the CESM-LE CAM variant uses the finite volume (FV) dynamical core. The dynamical core difference affects the dimensionality and layout of the output data, which impacts the effectiveness

of some compression algorithms. Thus, we started this study with no assumptions on what level of *fpzip* compression to use for each variable.

To determine a reasonable level of compression for each of the 159 CESM-LE CAM variables, we created a test ensemble of 101 12-month CESM simulations with a similar (but distinct) setup to the production CESM-LE simulations. Unlike the test ensemble in Baker et al. (2014), which only produced annual averages, we output daily, 6-hourly, and monthly data for the

simulation year and created ensembles for each frequency of output for each variable (212 total). We then used the size 101





test ensemble to chose the *fpzip* parameters that yielded the lowest CR such that the suite of four quality metrics proposed in Baker et al. (2014) all passed. Note that we did not use CESM-LE members 1-30 for guidance when setting the *fpzip* precision parameters for compressing the two new ensemble runs, but based all selections on the variability of the size 101 test ensemble. (Note that an ensemble with 101 has more variability than one with 30 members) Finally, we mention that several variables

occasionally contain "missing" values (i.e., there is no data value at a grid point). While "fill" values (i.e., a defined fixed value to represent missing data) can be handled by *fpzip*, it cannot process the locations with missing data (which would need to be either populated with a fill value or masked out in a preprocessing step). Therefore the following CESM-LE variables are not compressed at all: TOT_CLD_VISTAU, ABSORB, EXTINCT, PHIS, SOLIN, AODDUST2, LANDFRAC, and SFCO2_FFF.

The first two rows in Table 1 list the compression ratios for each of the output frequencies for both *fpzip* and the lossless

compression that is part of the NetCDF-4 library (*zlib*). Note that applying the customized-by-variable *fpzip* parameters to a single CESM-LE ensemble member (180 simulation years) yielded an average CR of 0.18 (more than a 5:1 reduction), which is a 3.5x reduction over the lossless NetCDF4 library compression. The third row in Table 1, labeled "truncation", indicates the compression ratios possible with simple truncation if each variable was truncated to the same precision as specified for *fpzip*. (Table 2 lists how many variables out of the 212 total used each level of *fpzip* compression). Therefore, the differences

between the compression ratios for *fpzip* and truncation in Table 1 highlight the added value of *fpzip*'s predictor and encoder in reducing data volumes over simple truncation. Note that Table 2 shows that the majority of the variables were able to use the most agressive compression, *fpzip-16*.

## 4   Ensemble data evaluations

In this section, we describe selected analyses performed on the CESM-LE data that were conducted without prior knowledge

of which of the new ensemble members (31-33) had been subjected to lossy compression. These experiments were designed to identify which new ensemble members had been compressed and reconstructed and to determine whether the compression-induced effects were significant. Note that because *fpzip* truncates values (and is therefore biased towards zero), one could trivially compare the raw data files directly to determine which ensemble members had undergone compression and reconstruction. However, analyses in this section and the next look for data discrepancies via various methods typically applied in

climate analysis.

### 4.1   CVDP

We first discuss results from the Climate Variability Diagnostic Package (CVDP) (Phillips et al., 2014), a publicly available analysis tool for examining major modes of climate variability. In particular, the CVDP outputs a variety of key climate metrics, which are immediately viewable via a website of images (e.g., means, standard deviations, coupled modes of variability,

atmospheric modes of variability, global trend maps, AMOC, timeseries data, etc.). The CVDP was used to document the climate simulated by each member of the CESM-LE, and complete CVDP diagnostic data and images from several time periods are available on the CESM-LE project diagnostics page (http://www.cesm.ucar.edu/experiments/cesm1.1/LE/). Global





trend maps are one of the key metrics in the CVDP, and in Fig. 1, we show the CVDP-generated global trend map for annual air surface temperature (TAS) for historical simulation data (1920-2012). Note that this figure is comparable to Fig. 4 from Kay et al. (2015), but for annual data of a longer historical period. The three additional ensemble members (31-33) are shown in Fig. 1 as well. Also included are the reconstructed versions of 31 and 33, labeled 31-C and 33-C respectively. Note that there

is no discernible difference between 31 and 31-C and 33 and 33-C in this figure. This result is not unexpected as the types of calculations that the CVDP conducts are unlikely to identify compression effects. For that reason, all of the CVDP diagnostic data available on the CESM-LE project diagnostics page at present include the reconstructed variants of 31 and 33 (i.e., 31-C and 33-C in our figure) instead of the original uncompressed data (31 and 33 in our figure). No anomalies or differences have been reported for any of the CVDP diagnostic data for the CESM-LE project that include the reconstructed members 31 and

10    33.

## 4.2 Climate characteristics

We now describe an analysis aimed at determining whether the effects of the lossy compression could be distinguished from the internal variability inherent in the climate model as illustrated by the CESM-LE project ensemble member spread. The CESM-LE historical simulation (1920-2005) data is examined for ensemble members 2-33 (member 1 is excluded due to a

technicality related to its different starting date). Multiple characteristics of interest across the ensemble are examined: surface temperature, top of the atmosphere model radiation, surface energy balance, precipitation and evaporation, and differenced temperature fields. The effects of compression are discernable in several characteristics.

### 4.2.1 Surface temperature

First, we plot the global mean annual surface temperature evolution in Fig. 2. The three additional members (31-33) are within

the range of internal variability, and it cannot be determined which new member(s) has been compressed and reconstructed. Second, we examine the extreme values for surface temperature due to the often cited concern that applying compression to scientific data could dampen the extremes. We calculate the difference between the maximum monthly average and minimum monthly average surface temperature in 3-year segments. While the temperature difference was lowest for member 32 (which was not compressed) in the first 6 years, this trend did not continue through the remaining 80 years. In fact, none of the

members 31-33 show any detectable surface temperature anomalies as compared to the rest of the ensemble members.

### 4.2.2 Top of the atmosphere model radiation

Examining the top of the atmosphere (TOA) model radiation balance is of interest as compression could potentially violate conservation of mass, energy or momentum. TOA imbalance is calculated as net shortwave (SW) radiation minus the net longwave (LW) radiation. We found no discernable difference in the TOA radiation imbalance due to compression (that could

be distinguished from the ensemble variability) when we looked at members 1-33 in the time period 1920-2005 or the shorter period from 1920-1940, shown in Fig. 3. Furthermore, the TOA radiation imbalance timeseries in Fig. 4 also indicates that



internal variability is masking any possible effects due to compression. Note that we also examined the top of the model net LW and net SW radiation independently and that data did not indicate any anomalies in the new members either.

### 4.2.3 Surface energy balance

Surface energy balance is another popular climate model characteristic that is commonly calculated in climate model diagnostics. The energy balance at the Earth's surface indicates the heat storage in the climate system and is calculated as the sum of the net solar flux at the surface (FSNS), the net longwave flux at the surface (FLNS), the surface latent heat flux (LHFLX), and surface sensible heat flux (SHFLX) (e.g., see Raschke and Ohmura (2005)). We calculated the imbalance in the surface energy for each month using the monthly average output of variables FSNS, FLNS, LHFLX, and SHFLX. Fig. 5 shows the mean imbalance over the period from 1920-2005. Note that members 31 and 33 (both of which were compressed) fall far outside the range of internal variability. We found that the difference in surface energy balance for 31 and 33 is attributable to lower levels of the surface latent heat flux (LHFLX) for the reconstructed members, as seen in Fig. 6. We note that this larger surface energy imbalance persists in the later CESM-LE sets from 2006-2080.

We examined the four CESM-LE variables involved in the surface energy balance calculation. We found that LHFLX was compressed more aggressively than the other three variables (fpzip-16 versus fpzip-24). Therefore, we repeated the surface energy balance calculation with LHFLX subjected to fpzip24 (instead of fpzip16) and found that the surface energy balance anomalies for members 31 and 33 disappear. Fig. 7 shows the new result. Clearly relationships between variables can be important when determining an appropriate amount of compression to apply, especially in the context of derived variables. We further discuss this lesson in Sect. 6.

### 4.2.4 Precipitation and evaporation

Next we evaluated precipitation (the sum of variables PRECC and PRECL) across the ensemble, shown in Fig. 8, supposing that precipitation levels could be lower in 31 and 33 due to reduced surface latent heat flux (LHFLX); however, members 31 and 33 do not stand out in the context of precipitation. Evaporation, on the other hand, is directly calculated from latent heat flux (LHFLX) via a constant conversion factor (accounting for water density and latent heat of evaporation) that we determined from the first ensemble member (such that precipitation and evaporation were equal). A look at the evaporation across the ensemble showed lower levels of evaporation corresponding to members 31 and 33, resulting in the precipitation/evaporation imbalance shown in Fig. 9.

Both PRECC and PRECL were compressed with fpzip-24, whereas LHFLX used fpzip-16. As with the previously discussed surface energy balance calculation, the size of the anomalies in Fig. 9 points to the issue of a derived variable calculated from variables with differing levels of compression-induced error. Therefore, if we redo the precipitation/evaporation imbalance using LHFLX compressed with fpzip-24, the discrepancy between members 31 and 33 and the rest of the ensemble disappears, e.g. Fig. 10.





### 4.2.5 Differenced temperature field

Difference fields are useful for indicating whether key features of the field have been preserved (e.g. gradients). For each of the ensemble members, we calculate the difference field for the near-surface air temperature (TREFHT) field monthly mean for October 1920. In particular, we calculate the difference in near-surface air temperature between pairs of neighboring grid points first in longitudinal direction and then in latitudinal direction. Looking at the distribution of all these differences for each member via mean, median, interquartile range (IQR), and skewness, we found that reconstructed members were outliers only in terms of the IQR. The IQR is the third (upper) quartile minus the first (lower) quartile and indicates the spread of the distribution. Reconstructed members 31 and 33 have an IQR near 0.25, which is larger than that of any of the other ensemble members, which are all close to 0.1. Note that the median of the two reconstructed member's difference distributions is exactly zero. In fact, of all the differences calculated for each member, the value zero occurs less than one-tenth of a percent among the original members, but it occurs in about a third of the reconstructed ensemble members 31 and 33. This result means that neighboring values are often the same after compression, whereas they were not exactly the same originally. This detectable effect with a lossy method is expected as some precision has been lost. However, the difference is not necessarily relevant for analysis. For example, for temperature, one could argue that the last several digits were likely simulation noise and were, therefore, unimportant in terms of scientific conclusions. However, if compression dampens the minimum and maximum values that occur in the neighbor differences temperature field, this effect would be problematic. We calculated the difference between the minimum and maximum values in the differenced temperature field for each ensemble member for every month from 1920 to 2005. This calculation characterizes the largest temperature gradients that occur for each month. We show the October 1920 results in Fig. 11, which indicate that nothing is amiss with members 31 and 33 (which was also the case for all of the other months from 1920-2005).

In general, though, determining whether compression caused an overall smoothing effect on the data is perhaps better viewed by examining spatial contrast plots showing the North-South and East-West differences for the near-surface air temperature for the ensemble members. For ensemble member 31, Fig. 12 shows both the original (labeled "Member 31") and reconstructed (labeled "Member 31-C") data from October of 1920. Note that the scale of the color bar would need to be greater than +/-10 degrees to represent all gradients, but at that scale differentiating the smaller gradients is difficult and no compression effects can be detected. Therefore, the rightmost plots in Fig. 12 have a color bar scale tightly restricted to +/-.5 degrees. At this restricted scale, one can notice the effects of lossy compression largely over the ocean in areas where the original gradient was quite small already. However, when the color scale is slightly expanded to +/-1.5 degrees (in the leftmost plots), it is difficult to discern any differences between 31 and 31-C, and the larger gradients over land coastlines and ridges dominate, as expected.

### 4.3 Ensemble variability patterns

The idea behind the following analysis was to determine whether lossy compression would introduce detectable small scale variability patterns into the climate data. To this end, we reconstructed each large ensemble member (1-33) from a basis set derived from the variability from each other member of the large ensemble, with the idea that the complete basis set derived



from the compressed members would be able to explain less variance in the other simulations (because some of the higher modes would not be well-represented).

In particular, we followed the following procedure. For each ensemble member (1-33), we did a singular value decomposition (SVD) analysis to determine the EOFs (Empirical Orthogonal Functions) in the spatial dimension on the monthly temperature field for 900 months. Note that we examined a subset of the grid-cells to reduce computational costs. We then projected each of the remaining 32 ensemble members onto the resulting EOF basis and calculated the unexplained variance. Figure 13 provides the sum of the unexplained variance (mean-squared error) in temperature for each ensemble member (note that the expectation value has been subtracted for clarity). Figure 13 indicates that members 31 and 33 are outliers, meaning that their set of EOFs is less appropriate as a basis set to describe the variability in the other ensemble members; this is due to loss of precision induced by lossy compression (which primarily affects the high frequency modes).

Figure 14 shows the same result in an alternative way. Each subplot uses a set of EOFs (900 total) derived from a member of the large ensemble (subplots are only shown for members 21-33, as 1-20 share similar characteristics to the other members not subject to compression). The remaining 32 members are projected onto the EOF basis set, and we calculate the variance of the principal components in the rest of the ensemble (900 x 32 samples). The anomaly of this curve relative to the ensemble mean case is plotted in the subplots in Fig. 14. The subplot x-axes represent the 900 EOFs, and the y-axis indicates the magnitude of the temperature variance. The subplots for ensemble members 31 and 33 indicate that when the rest of the ensemble members projected onto their EOFs, those modes of rank 500 or greater exhibit lower than expected variance. Again, the reconstructed members do not contain the high frequency information present in the rest of the ensemble. Of note is that when we alternatively first derived EOFs in members 1-30 and then projected members 31 and 33 onto that basis set, no differences were detected as expected. Given that the differences are only noticeable in the higher EOFs (which are not typically examined), it appears that the compressed members are not (noticeably) under-representing any of the true modes of variability.

A natural question is whether the detected differences in variances for members 31 and 33 could impact science results. Clearly the large scale patterns of variability, long term trends, and regionally averaged properties would all be unaffected because they can be represented with a fraction of this number of EOF modes (i.e., many fewer than 500). Analyses that could potentially be affected by the truncation of EOF modes greater than 500 include such features as point-scale extreme temperatures or precipitation. We partially address this issue in Sect. 5.1 by investigating the extremes. However, in future work we will further explore whether compression-induced damping of high frequency elements (spatially or temporally) has relevant effects that exceed the noise stemming from the model's floating-point calculations.

### 4.4 Coherent structures

#### 4.4.1 Overview of Proper Orthogonal Decomposition

Proper Orthogonal Decomposition (POD) is used for the extraction of coherent structures, or the study of recurring patterns in spatio-temporal fields. The POD technique was first introduced in the context of fluid turbulence by Lumley (1967) in order to analyze the velocity field of fluid flows. POD has since been adapted for use within a number of different disciplines, such as



oceanography, chemistry, and model order reduction (Carbone et al., 2011). The aim of POD is to provide an optimal basis set to represent the dynamics of a spatio-temporal field, which allows the identification of the *essential information* contained in the signal by means of relatively few basis elements (modes).

In particular, given a spatio-temporal field $I(\boldsymbol{x},t)$, POD calculates a set of modes $\Phi$ in a certain Hilbert space adapted to the field $I(\boldsymbol{x},t)$ such that

$$I(\boldsymbol{x},t) = \sum_{i=1}^{\infty} a_i(t)\phi_i(\boldsymbol{x}), \tag{2}$$

where $a_i(t)$ is a time-varying coefficient. From a mathematical point of view, POD permits the maximization of the projection of the field $I(\boldsymbol{x},t)$ on $\Phi$:

$$\text{Max}_{\phi_i} \frac{\langle(I(\boldsymbol{x},t),\phi_i(\boldsymbol{x}))\rangle}{(\phi_i(\boldsymbol{x}),\phi_i(\boldsymbol{x}))}. \tag{3}$$

This defines a constrained Euler-Lagrange maximization problem, the solution of which is a Fredholm integral equation of the first kind:

$$\int_{\Omega} \langle(I(\boldsymbol{x},t),I(\boldsymbol{x}',t))\rangle\phi_i(\boldsymbol{x}')d\boldsymbol{x}' = \lambda_i\phi_i(\boldsymbol{x}), \tag{4}$$

where $(a,b)$ is the inner product, angle brackets indicate the time average, $\Omega$ is the spatial domain, $\phi_i(\boldsymbol{x})$ is an eigenfunction, and $\lambda_i$ is a real positive eigenvalue. If the spatial domain $\Omega$ is bounded, this decomposition provides a countable, infinite, set of sorted eigenvalues $\lambda_i$ (with $\lambda_1 \geq \lambda_2 \geq \lambda_3 \geq \ldots$). Then the field "energy", by the analogy with the fluid turbulence application, can be written as:

$$\langle I(\boldsymbol{x},t)\rangle = \sum_{n=1}^{\infty} \lambda_i, \tag{5}$$

where $\lambda_i$ represents the average energy of the system projected onto the axis $\phi_i(\boldsymbol{x})$ in the eigenfunction space. In general, the eigenfunction $\phi_i(\boldsymbol{x})$ does not depend on the functions of the linearized problem, but emerges directly from the observations of the field $I(\boldsymbol{x},t)$. When the sum in Equation (2) is truncated to $N$ terms, it contains the largest possible energy with respect to any other linear decomposition belonging to the family of EOFs (i.e. PCA, SVD) of the same truncation order (Lumley, 1967).

### 4.4.2 Application to ensemble data

For this study, we utilize POD to investigate whether lossy compression introduced any detectable artifacts that could indicate which ensemble member(s) of the new set 31-33 had been compressed and to determine whether any such artifacts were acceptable or not (i.e, in terms of impact on the physics of the problem). We examined the monthly averaged output of four variables: Z3 (geopotential height above sea level), CCN3 (cloud condensation nuclei concentration), U (zonal wind) and FSDSC (clear sky downwelling solar flux at surface). For each variable and for each ensemble member (1-33), POD was applied to a period of 25 years (300 time slices beginning with Jan. 2006) to obtain the modes and the energy associated





with each mode. This methodology enables the identification of any perturbations introduced by the compression method into the dynamics of the field. In addition, we can characterize the impact of the compression, if any, with respect to the inherent variability within the ensemble.

To illustrate this process, the "energy" fraction $\lambda$ as a function of the mode number $N$ is reported in Fig. 15 for variable Z3
from ensemble member 28. Note that the distribution of $\lambda$ is composed of different branches (groups of modes) characterized by a power-law behavior. The first branches (a, b and c) represent the dominant scales (structures) in the field and contain the greater part of the energy of the original field. These structures can be considered mother structures, and, analogously to the fluid turbulence case, they represent the "energy injection" point for the smaller structures. In other words, the large scale structures transfer energy to smaller and smaller scale structures. When a break is found in the distribution, the energy transfer
is stopped, and a new cascade begins that is unrelated to the previous one. For the highest modes ($\sim$scale of the resolution) the energy is quite low, and the modes have a minimal impact on the "physics". Beyond this point the modes can be considered uncorrelated noise, which is generally associated with the thermal noise of the floating point calculations, rather than anything physically meaningful. Therefore by comparing the energy distribution of the decomposition modes of the new ensemble members 31 -33 with their inherent variability, it should be possible to both identify the presence of any perturbations due to
the compression algorithm as well as the scale at which the perturbations are significant.

The four plots in Figure 16 correspond to each of the 4 variables analyzed. First, panel (a) shows the energy distribution of the modes of the POD of new ensemble members 31-33, superimposed on the median of the original ensemble members (1-30). To highlight the differences of the energy distributions of the modes of the decomposition of members 31-33 with respect to the median of the original ensemble members, their relative errors are reported in panel (b) together with the natural
variability observed within the original ensemble. Finally, panel (c) reports the distribution of the root-mean-square Z-score (RMSZ) of the energy distribution for the original ensemble members together with the RMSZ of the energy distribution of members 31-33. The plot corresponding to variable Z3 in Figure 16 clearly shows that the RMSZ values for members 31 and 33 are outliers in panel (c), suggesting that there are some artifacts in the distribution energy of the modes of their relative PODs, potentially caused by lossy compression. However, when comparing these errors with the natural variability observed
within the original ensemble, it appears clear that such anomalies are mainly visible in the lowest energy modes ($> 150$). Since the lowest energy modes are generally attributed to thermal noise in floating point calculations, if these artifacts are due to lossy compression, they do not affect any coherent structures attributable to the physics of the problem (i.e., the climate). Note that ensemble members 31 and 33 for variables U and FSDSC exhibit errors in the energy distribution that in a few instances exceed the natural variability within the ensemble as shown in panel (b), but the exceedence is not great enough to clearly
indicate them as outliers in panel (c). (Recall that ensemble member 32 was not compressed.) Finally, the errors in the energy distribution of the modes of the decomposition for ensemble members 31-33 for variable CCN3 are well within the variation explained by the natural variability of the original ensemble members, and therefore no outliers were observed.

This analysis performed on a limited number of variables shows that the compression of the ensemble members has either no effect if compared with the natural variability observed within the ensemble, or (for Z3) affects only the lowest energy modes.



We note that the outcome of this analysis could potentially be different if applied to higher temporal resolution output data as lossy compression could impact finer scale patterns differently.

## 5    The original and reconstructed data

In this section, we describe analyses performed on the CESM-LE data that were conducted with the knowledge that members 31 and 33 had been compressed and reconstructed. In addition, we provided both the original and reconstructed versions of 31 and 33 for these experiments.

### 5.1    Climate extremes

#### 5.1.1    Overview of extreme value theory

Extreme value theory, as the name implies, focuses on extremes, more precisely on the upper tail distributional features. For extremes, the Gaussian paradigm is not applicable. To see this, suppose that we are interested in annual maxima of daily precipitation. In this case, the probability density function (pdf) is skewed, bounded by zero on the left side, and very large values (greater than two standard deviations) can be frequently observed. These three features cannot be captured by a normal distribution and other statistical modeling tools are needed.

One classical approach is to study block maxima, e.g., the largest annual value of daily temperatures. In this example, the block size is 365 days. The statistical analysis of block maxima is based on the well- developed extreme value theory (EVT), originating from the pioneering work of Fisher and Tippett (1928) and regularly improved upon during the last decades (e.g., De Haan and Ferreira (2005)). This theory indicates that the generalized extreme value distribution (GEV) represents the ideal candidate for modeling the marginal distribution of block maxima. This probabilistic framework is frequently applied in climate and hydrological studies dealing with extremes (e.g., Zwiers et al. (2013), Katz et al. (2002)). Nowadays, more complex statistical models, such as the multivariate EVT (e.g., De Haan and Ferreira (2005), Beirlant et al. (2004), Embrechts et al. (1997)), also provide a theoretical blueprint to represent dependencies among maxima recorded at different locations. For this work, however, we won't address the question of spatial dependencies for extremes. We assume that every grid point can be treated independently and a GEV can be fitted at each location.

Mathematically, the GEV is defined by its three-parameter cumulative distribution function (cdf):

$$G(y) = \exp\left(-\left(1 + \xi \frac{y-\mu}{\sigma}\right)_+^{-1/\xi}\right), \tag{6}$$

where $\mu$, $\sigma > 0$ and $\xi$ are called the location, scale and shape parameter with the constraint that $1 + \xi \frac{y-\mu}{\sigma} > 0$. The $\xi$ parameter defines the tail behavior with three possible types: $\xi = 0$ (Gumbel), $\xi > 0$ (Fréchet) and $\xi < 0$ (Weibull). Temperature extremes often follow a Weibull distribution (e.g., Zwiers et al. (2013)). In particular, a negative shape parameter implies a finite upper bound given by $\mu - \frac{\sigma}{\xi}$. For other examples in environmental area, the Gumbel family has been used to model daily maxima of methane (Toulemonde et al., 2013) and precipitation maxima are often described by a Fréchet distribution (see, e.g., Cooley



et al. (2007)). In terms of risk analysis, the scalar $\xi$ is the most important parameter of the GEV parameters. For this reason, most of our analysis will be based on assessing if and how $\xi$ changes with compression.

### 5.1.2 Application to ensemble data

We focus our analysis on four variables from the ensemble data: average convective and large-scale precipitation rate (PRECT)
over the output period, maximum convective and large-scale precipitation rate (PRECTMX), minimum surface temperature over output period (TSMIN), and maximum surface temperature over output period (TSMAX). We study TSMX, PRECTMX and PRECT using annual block maxima, and TSMN using annual block minima (the GEV can be applied by multiplying by $-1$). Concerning the inference of the GEV parameters, most classical approaches, including MLE (maximum likelihood estimation), MOM (method of moments), and Bayesian methods, can be used. As the shape parameter for precipitation and
temperatures extremes is classically between $-.5$ and $.5$, we opt for the so-called Probability Weighted Moments (PWM) (e.g., Ana and de Haan (2015)), which has a long tradition in statistical hydrology (e.g., Landwehr et al. (1979), Hosking and Wallis (1987)) and has been applied in various settings (e.g., Toreti et al. (2014)). Besides its simplicity, the PWMs approach usually performs reasonably well compared to other estimation procedures (e.g., Caeiro and Gomes (2011)). Additional arguments in favor of PWMs are that they are typically quickly computed, an important feature in our setup, and do not provide aberrant
values for negative $\xi$ like the MLE. To apply this estimation technique to temperature min and max, global warming trends have to be removed. This was done by removing the trend with a local non-parametric regression (using the *loess* function in R).

Figure 17 summarizes our findings concerning the shape parameter $\xi$. Each row represents a variable of interest. We only show results for one of the two compressed ensemble members as the results are practically identical. The histograms corre-
spond to the empirical pdf obtained from all uncompressed runs. This can be compared to the blue pdf of the compressed run. For our four atmospheric variables, one cannot make the distinction between the compressed and uncompressed runs, which indicates that compression did not systematically change the distribution of the shape parameters. The middle panels display the range of the estimated $\xi$ at each grid point from the ensemble of 31 uncompressed runs. This gives us information on the variability among the 31 uncompressed runs, which can be compared to the difference between a compressed run and its un-
compressed counterpart (the right panels). As indicated by the dark blue color (meaning low values), the ensemble variability is much higher than the variability due to compression. In summary, this analysis indicates that compression does not cause any systematic change in the distribution of the estimated shape parameters and that the changes introduced by compression fall well within the variability of the ensemble.

### 5.2 Causal signatures

The goal of causal discovery in this context is to identify potential cause-effect relationships from a dataset to better understand or discover the dynamic processes at work in a system. Causal discovery tools have been developed from probabilistic graphical models (e.g., detailed in Pearl (1988) and Spirtes et al. (2000)), which are a graphical representation of probable dependencies between variables in high dimensional space. In particular, causal discovery methods reveal more than simply correlation,



but rather the patterns of information flow and interactions. To determine the flow of information, the initial assumption is made that every variable (graph node) is causally connected to every other variable. Then conditional independence tests (e.g. testing for vanishing partial correlations) are used to disprove causal connections, resulting in a remaining "interaction map" of causal connections (that may or may not be given direction through additional techniques). Such tools were initially applied

in the fields of social sciences and economics, but have more recently been applied successfully to climate science data (e.g., Chu et al. (2005), Ebert-Uphoff and Deng (2012a), Ebert-Uphoff and Deng (2012b), Zerenner et al. (2014)). For example, for atmospheric data, one could imagine using causal discovery methods to understand large-scale atmospheric processes in terms of information flow around the earth.

Of interest here is determining whether compressing the climate data in the CESM-LE dataset affected the flow of informa-

tion. Using causal discovery for this purpose is proposed in Hammerling et al. (2015), where interaction maps were generated for both the original and reconstructed data. We call these interaction maps causal signatures. This type of analysis is unique to this compression study as it is aimed at inter-variable relationships. Recall that the number of daily variables contained in the CESM-LE datasets is 51. To simplify the analysis, we created a subset of 15 daily variables. The subset was chosen such that only one variable was kept from each like-variable group. For example, eight of the 51 total daily variables report temperature

in some form: at several defined pressure surfaces, at the surface, and at a near-surface reference height (TREFHT), and, therefore, we only include the temperature variable TREFHT in the subset. We then developed temporal interaction maps for the 15 daily variables that show interactions across different lag times between variables. We performed this analysis for several different temporal scales, i.e. we identified separate signatures considering lag times between variables that are multiples of 1, 5, 10, 20, 30, or 60 days, in order to capture interactions for example on a daily (1 day) or monthly (30 days) scale. Recall that

these interaction maps are highlighting potential cause and effect relationships. Figure 18 contains the interaction map for the daily time scale (lag times are multiples of one day) for the original data for CESM-LE member 31, and the 15 variables are indicated in the ovals. Note that only the weak connection between SHFLX (surface sensible heat flux) and FSNTOA (net solar flux at top of atmosphere), which is indicated by a dotted line, is missing in the map corresponding to the reconstructed data. In general, the maps for all of the lagged times only indicated *tiny* differences between the initial and reconstructed datasets.

This result indicates that compressing and reconstructing the climate data has not negatively impacted the flow of information in terms of detectable cause-effect relationships in the data.

## 5.3  AMWG diagnostics package

The publicly available and popular AMWG (Atmosphere Working Group) Diagnostics Package (AMWG-DP) computes climatological means of CESM simulation data from CAM and produces plots and tables of the mean climate in a variety of

formats. The AMWG-DP uses monthly output to evaluate climate characteristics such as seasonal cycles, intraseasonal variability, Madden-Julian Oscillation (MJO), El Nino-Southern Oscillation (ENSO), and the diurnal cycle. The AMWG-DP can be used to compare model simulation output of observational and reanalysis data or to compare output from two simulations. Therefore, comparing the compressed and reconstructed CESM-LE ensemble members via the AMWG-DP is a natural choice. Note that the AWMG-DP is available at https://www2.cesm.ucar.edu/working-groups/amwg/amwg-diagnostics-package.



Because the AMWG-DP produces over 600 tables and plots, we just highlight a couple of results here. First we show vertical contour plots produced by the AMWG-DP (from Diagnostics Set 4) comparing the original and reconstructed variants of ensemble member 31 for relative humidity (RELHUM) in Fig. 19. We chose to look at RELHUM because it was compressed aggressively with fpzip-16, yielding a compression ratio (CR) of 0.09. While the max values are not identical (101.66 versus 101.86), the contour plots certainly appear very similar at this scale.

Now we show surface pressure (PS), as it is a "popular" variable to view with the AMWG-DP. Variable PS was compressed with fpzip-20, yielding a CR of 0.13. Figure 20 compares the original and reconstructed variants of ensemble member 31 via horizontal contour plots (from Diagnostics Set 5). Note that while the mean, max, and min values differ slightly, the plots themselves are indistinguishable and similar conclusions could be drawn.

Finally, we look at a portion of one of the AMWG-DP tables for global annual means for the 2006-2099 data (from Diagnostics Set 1) shown in Table 3. In particular, the AMWG-DP derived variables RESTOM and RESSURF are important diagnostics as they indicate the top of model residual energy balance and the surface residual energy balance, respectively, and Table 3 indicates that their computed values for the compressed and original cases are identical (to the precision used by AMWG-DP). Recall that top of the model energy imbalance was examined in Sect. 4.2.2 and is simply the difference between the net solar flux (shortwave radiation) at the top of the model (FSNT) and the net longwave flux at the top of the model (FLNT). Monthly variables FSNT and FLNT were both compressed by fpzip-24. The AMWG-DP results for RESTOM agree with the findings in Section 4.2.2 that indicate that the compression error cannot be distinguished from the ensemble variability. However, this simple diagnostic calculation warrants further discussion. We note that compressing FSNT with fpzip-16 and FLNT with fpzip-20 was acceptable in terms of passing the four quality metrics used to determine compression levels (see discussion in Section 3). However, because we knew in advance of applying compression that calculating the top of the model balance (FSNT - FSLT) is a key diagnostic check for climate scientists, we preemptively used less aggressive compression for both variables (as subtracting like-sized quantities would magnify the error due to compression). For example, had we instead used FSNT with fpzip-16 and FLNT with fpzip-20, this would have resulted in relative errors of $0.3\%$ and $0.02\%$ for FSNT and FLNT respectively, but in a relative error for the derived quantity RESTOM of $8.0\%$, which is noticeably larger (corresponding to RESTOM values of $7.553$ and $8.211 W/m^2$)

The AWMG-DP derived quantity RESSURF for surface residual energy balance in Table 3 is notably on target in the compressed data. In contrast, when the surface energy balance was investigated in Sect. 4.2.3, Fig. 5 indicated that the effects of compression were noticeable in the surface energy calculation (due to aggressive compression of the surface latent heat flux, LHFLX). In both Sect. 4.2.3 and AMWG-DP, the surface energy balance was calculated as (FSNS - FLNS - SHFLX - LHFLX). However, the difference is that the AMWG-DP does not use the LHFLX variable from the output data, but instead calculates surface latent heat flux via surface water flux (QFLX) and four precipitation variables (PRECC, PRECL, PRECSC, and PRECSL). As a result, compression of variable LHFLX did not affect the AMWG-DP's calculation of surface energy balance.





## 6 Lessons learned

By providing climate scientists with access to data that had undergone lossy compression, we received valuable feedback and insights into the practicalities of applying data compression to a large climate dataset. Here we summarize the underlying themes or lessons that we learned from this lossy compression evaluation activity.

### 6.1 Relationships between variables

When determining appropriate levels of compression, relationships between variables can be an important consideration, particularly in the context of derived variables. As an example, we refer to the surface energy balance anomaly detected and discussed in Sect. 4.2.3. Had all four variables been compressed to the same precision, the surface energy balance in the reconstructed members would not have stood out (i.e., Fig. 5 versus Fig. 7). Derived variables are quite popular in post-processing analysis, and it is unrealistic to expect to know how the output data will be used at the time it is generated (and compressed). However, many derived variable calculations are quite standard (e.g., surface energy balance, top of the atmosphere energy balance, etc.), and these often-computed derived variables should be considered when determining appropriate levels of compression for variables used in their calculations.

### 6.2 Detectable versus consequential

A skilled researcher would likely be able to detect effects of lossy compression on data. However, the fact that the compression effects are detectable does not mean that they are also relevant and/or important. Recall that CESM model calculations happen in 64-bit precision, but the history files for post-processing analysis are output in 32-bit precision. Certainly there is no reason to believe that 32 bits are of consequence for every variable, and for many variables the trailing digits are model noise that would not impact scientific conclusions drawn from the data. For example, one would not expect to need 32-bits of precision to look at temperature and detect a warming trend. On the other hand, one may not want to study high-frequency scale events such as precipitation with data that has undergone aggressive compression. In general, understanding the precision and limitations of the data being used is critical to any post-processing analysis.

### 6.3 Individual treatment of variables

We confirmed the assertion in Baker et al. (2014) that determining the appropriate amount of compression must be done on a variable-by-variable basis. In particular, there is not a "one-size-fits-all" approach to compressing climate simulation variables, and it does not make sense to assume that 32 bits is the right precision for every variable. Further, customizing compression per variable could also include applying other types of compression algorithms to different variables as well (e.g. transform-based methods such as wavelets), which is a topic of future study. Knowing what precision is needed for each variable for CESM, or even more generally for CMIP (discussed in Sect. 1), would clearly facilitate applying lossy compression. We note that defining such a standard is non-trivial and would need to be fluid enough to accommodate new variables and time/space resolutions.





### 6.4 Implications for compression algorithms

Achieving the best compression ratio without negatively impacting the climate simulation data benefits from a thorough understanding how a particular algorithm achieves compression. For example, we are aware that the type of loss introduced by *fpzip* is of the exact same kind that is already applied to the original double-precision (64-bit) data when truncating (or, more commonly, rounding) to single precision (32-bit) for the CESM history file. Because of its truncation approach, *fpzip* is much less likely to affect extreme values or have smoothing effect on the data, as opposed to, for example, a tranform-based approach.

Further, Fig. 6 illustrates that naive truncation is not ideal. An improvement would be to inject random bits or at least round rather than truncate the values (i.e., append bits $100\ldots0$ instead of bits $000\ldots0$ to the truncated floats). Both of these modifications could be done as a post-processing step after the data have been reconstructed. Although the temperature gradients (as shown in Fig. 11) are not problematic in this study, injecting random bits would also reduce the number of zero gradients. On a related note, a compression algorithm that provides information about the compression error at each grid point could potentially be very useful in terms of customizing how aggressively to compress particular climate simulation variables.

Finally, an important issue for climate data is the need for compression algorithms to seamlessly handle both missing values and fill values. As mentioned previously, variables that occasionally have no value at all (i.e., missing) at seemingly random grid points require special handling by the compression algorithm itself or in a pre- and/or post-processing step. Similarly, the non-regular presence of large-magnitude fill values (typically $\mathcal{O}(10^{35})$ in CESM) can be problematic as well.

### 7 Concluding remarks

In general, lossy data compression can effectively reduce climate simulation data volumes without negatively impacting scientific conclusions. However, by providing climate researchers with access to a large dataset that had undergone compression (and soliciting feedback), we now better appreciate the complexity of this task. All of the lessons detailed in the previous section highlight the importance of being data- and science-aware when applying data compression and performing data-analysis. To reap the most benefit in terms of achieving low compression ratios without introducing statistically significant data effects requires an understanding of the characteristics of the data, their science use, and the properties (i.e., strengths and weaknesses) of the compression algorithm. Further, our compression research thus far has focused on evaluating individual variables, and this study highlights that issues can arise when compressing multiple variables or using derived variables. Our ongoing research on compression methods will focus on incorporating the multivariate aspects of compression and ultimately developing a tool to auto-determine appropriate compression (and therefore acceptable precision) for a given variable.

### 8 Code and data availability

All data for the CESM-LE project (including compressed and reconstructed members 31 an 33) are available via the Earth System Grid (http://www.earthsystemgrid.org). Also see https://www2.cesm.ucar.edu/models/experiments/LENS for more detailed information on the CESM-LE data. The CESM software is available from http://www2.cesm.ucar.edu/model, and



CESM-LE data was generated with the CAM5 configuration described in Kay et al. (2015). The *fpzip* compression utility is available from https://computation.llnl.gov/casc/fpzip/.

## Appendix A: Lossy compression evaluations

Table 4 lists which co-authors conducted the ensemble data evaluations described in Sections 4 and 5.

5 *Acknowledgements.* We thank Adam Phillips (NCAR) for his input on CVDP. We thank Reto Knutti (ETH Zurich) for his suggestions and ideas. We also thank William Kaufman (Fairview HS) for his work on Fig. 17. This research used computing resources provided by the Climate Simulation Laboratory at NCAR's Computational and Information Systems Laboratory (CISL), sponsored by the National Science Foundation and other agencies. Part of this work has been supported by the ANR-DADA, LEFE-INSU-Multirisk, AMERISKA, A2C2 and Extremoscope projects. Part of the work was done during the visit of P. Naveau at IMAGe-NCAR in Boulder, CO, USA. Part of this work

10 was performed under the auspices of the U.S. Department of Energy by Lawrence Livermore National Laboratory under Contract DE-AC52-07NA27344, and this material is based upon work supported by the U.S. Department of Energy, Office of Science, Office of Advanced Scientific Computing Research.



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





**Table 1.** Impact in terms of compression ratios (CR) of lossy compression with *fpzip*, lossless compression with *NetCDF-4*, and simple truncation for a CESM-LE ensemble member.

| Method | Monthly | Daily | 6-hourly | Average |
|---|---|---|---|---|
| *fpzip* | .15 | .22 | .18 | .18 |
| *NetCDF-4* | .51 | .70 | .63 | .62 |
| truncation | .61 | .58 | .60 | .69 |

**Table 2.** The number of variables that used each *fpzip* compression level (in terms of number of bits retained). Note that NC means "not compressed" due to missing values.

| Number of bits retained | 16 | 20 | 24 | 28 | 32 | NC |
|---|---|---|---|---|---|---|
| Monthly variable | 75 | 31 | 15 | 1 | 6 | 8 |
| Daily variables | 29 | 11 | 11 | 0 | 0 | 0 |
| 6-hourly variables | 12 | 8 | 4 | 0 | 0 | 1 |
| Total | 116 | 50 | 30 | 1 | 6 | 9 |





**Figure 1.** CVDP-generated global maps of historical (1920-2012) annual surface air temperature trends for the 30 original individual CESM-LE ensembles member, the three new members (31-33), and the reconstructed data from new members 31 and 33 (contained in the lower right box).





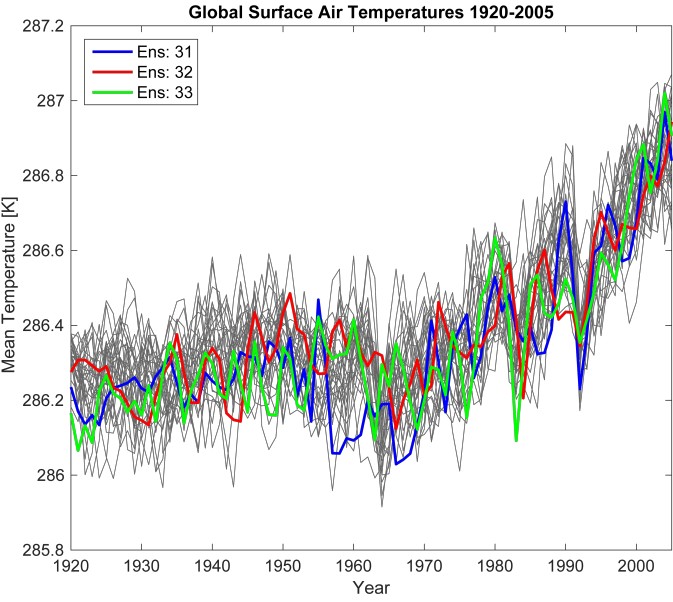

**Figure 2.** Annual global mean surface temperature evolution for 1920-2005. CESM-LE members 2-30 are indicted in gray and the three new members (31-33) are designated in the legend.

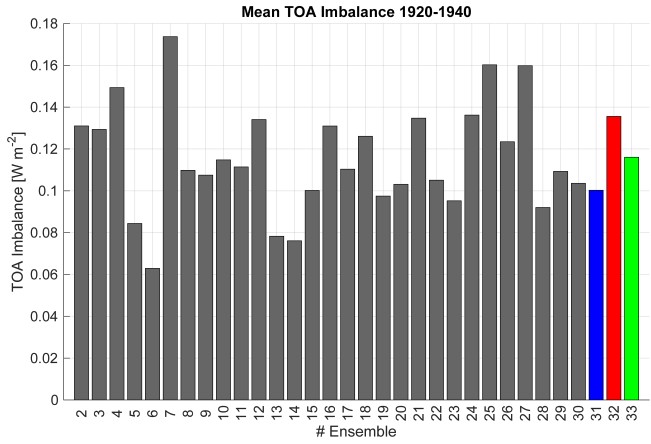

**Figure 3.** Global mean of top of model energy imbalance for 1920-1940 for CESM-LE members 2-30 and the three new members (31-33).



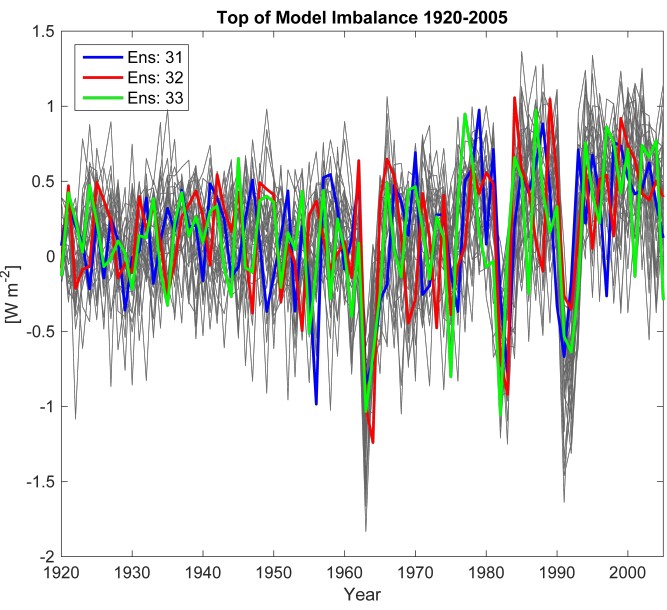

**Figure 4.** Top of model energy imbalance for 1920-2005. CESM-LE members 2-30 are indicted in gray and the three new members (31-33) are designated in the legend.

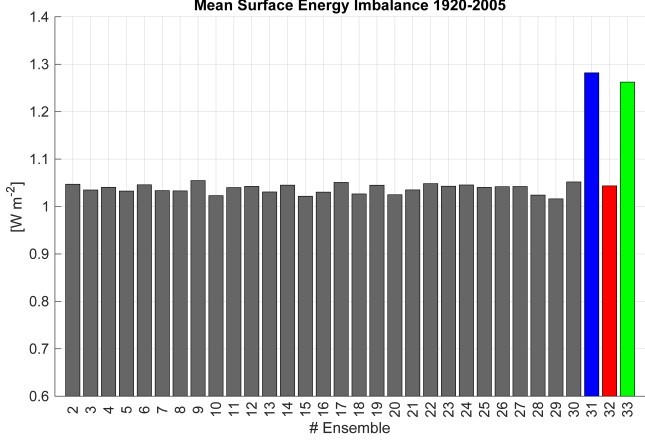

**Figure 5.** Mean surface energy imbalance for 1920-2005 for CESM-LE members 2-30 and new members 31-33.





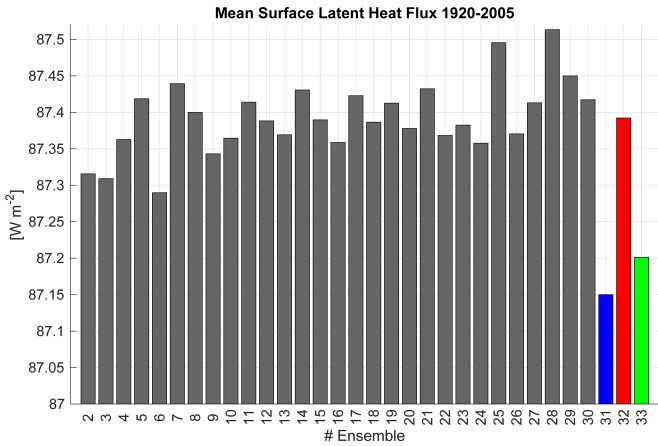

**Figure 6.** Mean surface latent heat flux (LHFLX) for 1920-2005 for CESM-LE members 2-30 and new members 31-33.

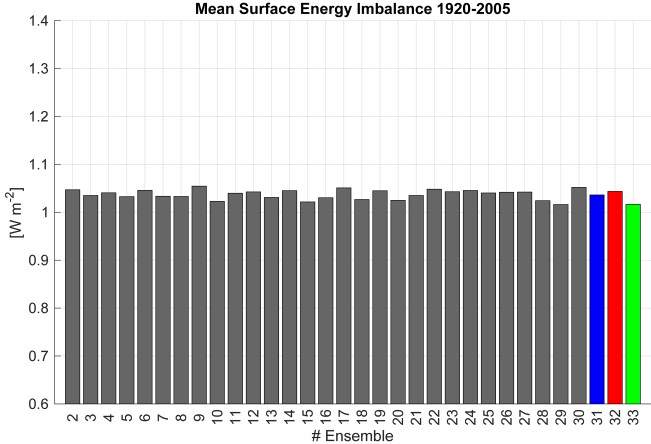

**Figure 7.** Mean surface energy imbalance for 1920-2005 for CESM-LE members 2-30 and new members 31-33 with adjusted compression level for LHFLX.





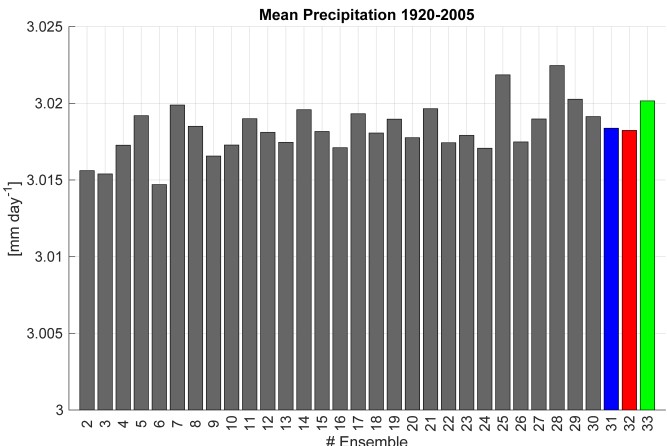

**Figure 8.** Mean precipitation for 1920-2005 for CESM-LE members 2-30 and new members 31-33.

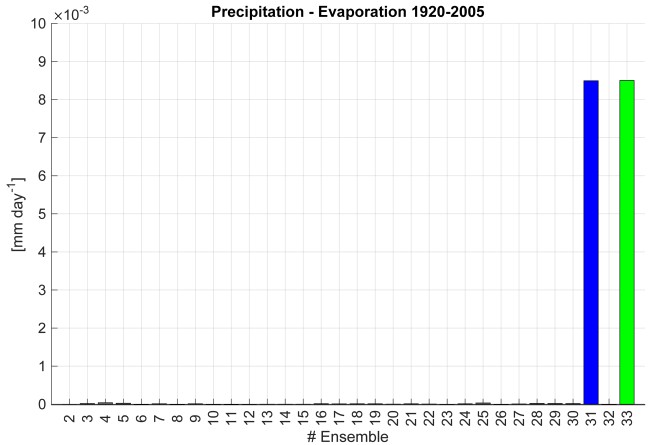

**Figure 9.** The balance between precipitation and evaporation for 1920-2005 for CESM-LE members 2-30 and new members 31-33.





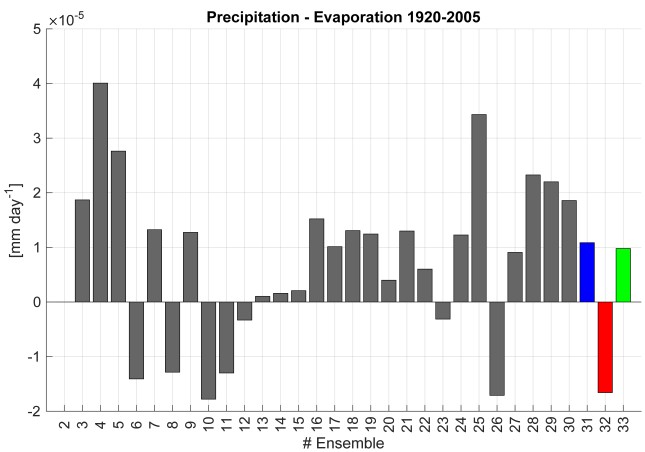

**Figure 10.** The balance between precipitation and evaporation for 1920-2005 for CESM-LE members 2-30 and new members 31-33 with adjusted compression level for LHFLX. Note the difference in scale between this plot and that in Figure 9.

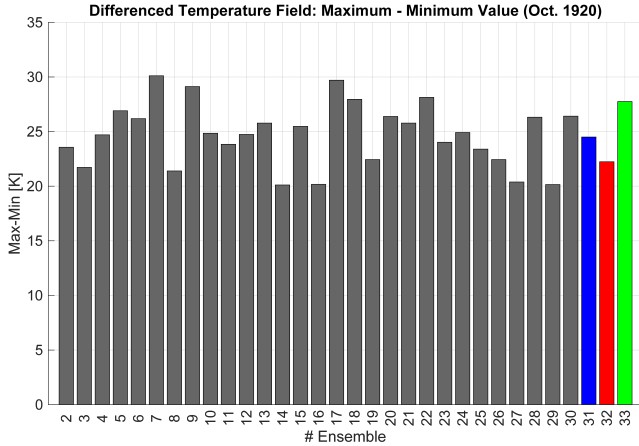

**Figure 11.** Difference between maximum and minimum values occurring in the neighbor differences surface temperature field (TREFHT) for each ensemble member for October 1920.




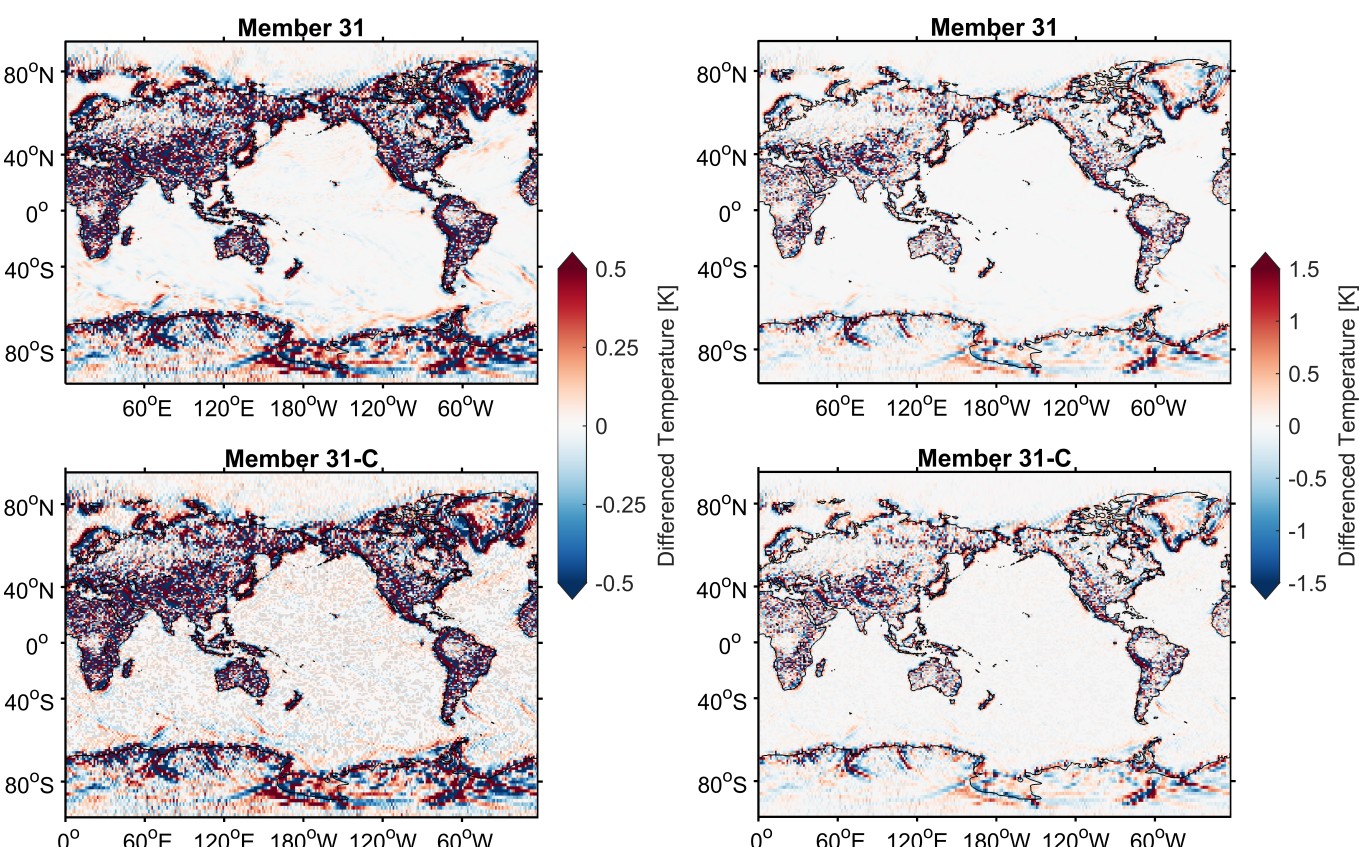

**Figure 12.** A comparison of the difference maps (i.e., gradients) for the surface temperature field (TREFHT) for ensemble members 31 (original) and 31-C (reconstructed) for October 1920. Note that the color scale for the left maps has a smaller range than for the right maps.




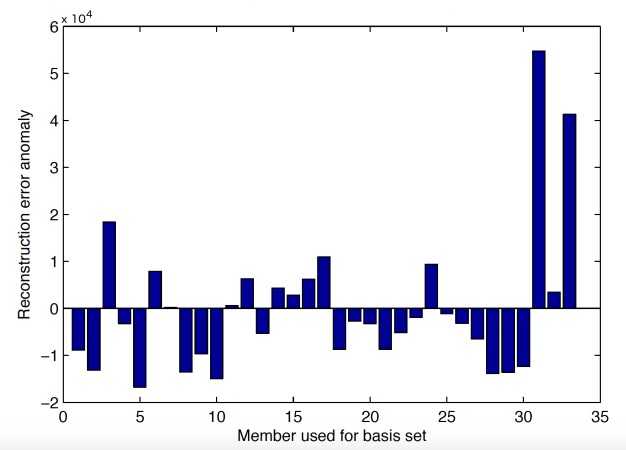

**Figure 13.** The sum of the mean-squared error in temperature field when the other ensemble members' variance is projected onto a single member's EOF basis.

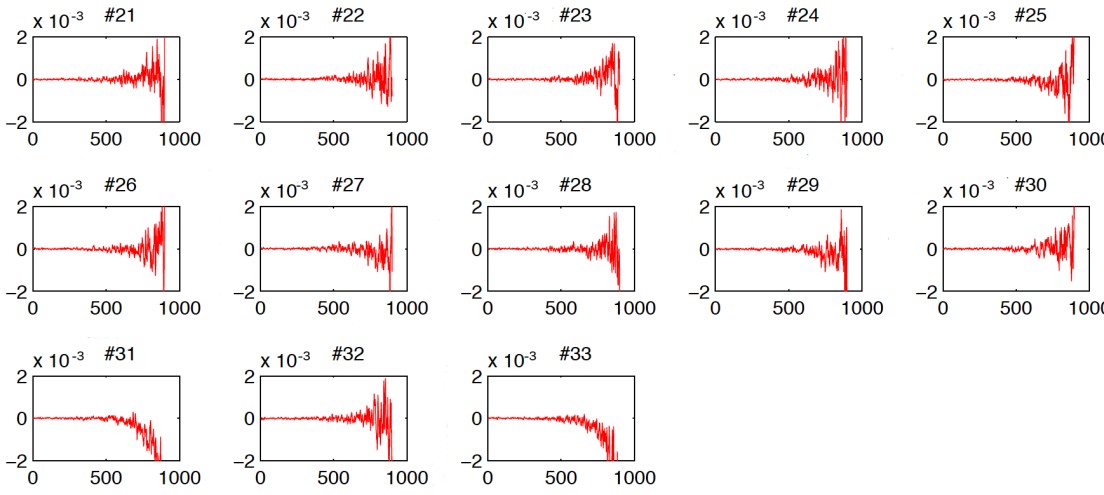

**Figure 14.** The subplot x-axes represent the 900 EOFs. The y-axes indicate the magnitude of the temperature variance. The ensemble member number is indicated in each subplot title, and members 31 and 33 have been subjected to lossy compression





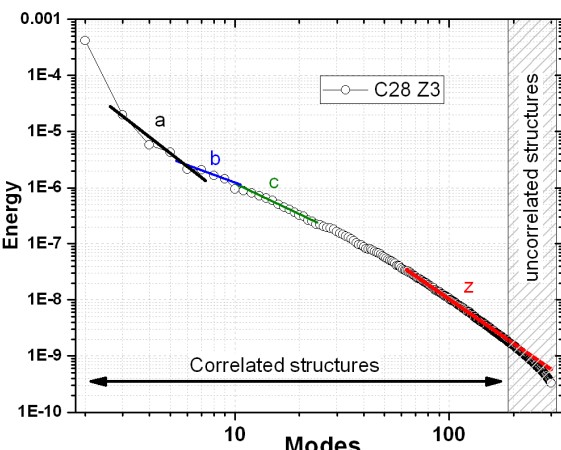

**Figure 15.** Energy distribution of the modes of the POD for variable Z3 for ensemble member 28. Superimposed power laws indicate the "energy cascades" in correlated modes, and three principal scales are present: a, b and c. The limit of the cascade is labeled z, and the shaded area indicates modes associated with noise.





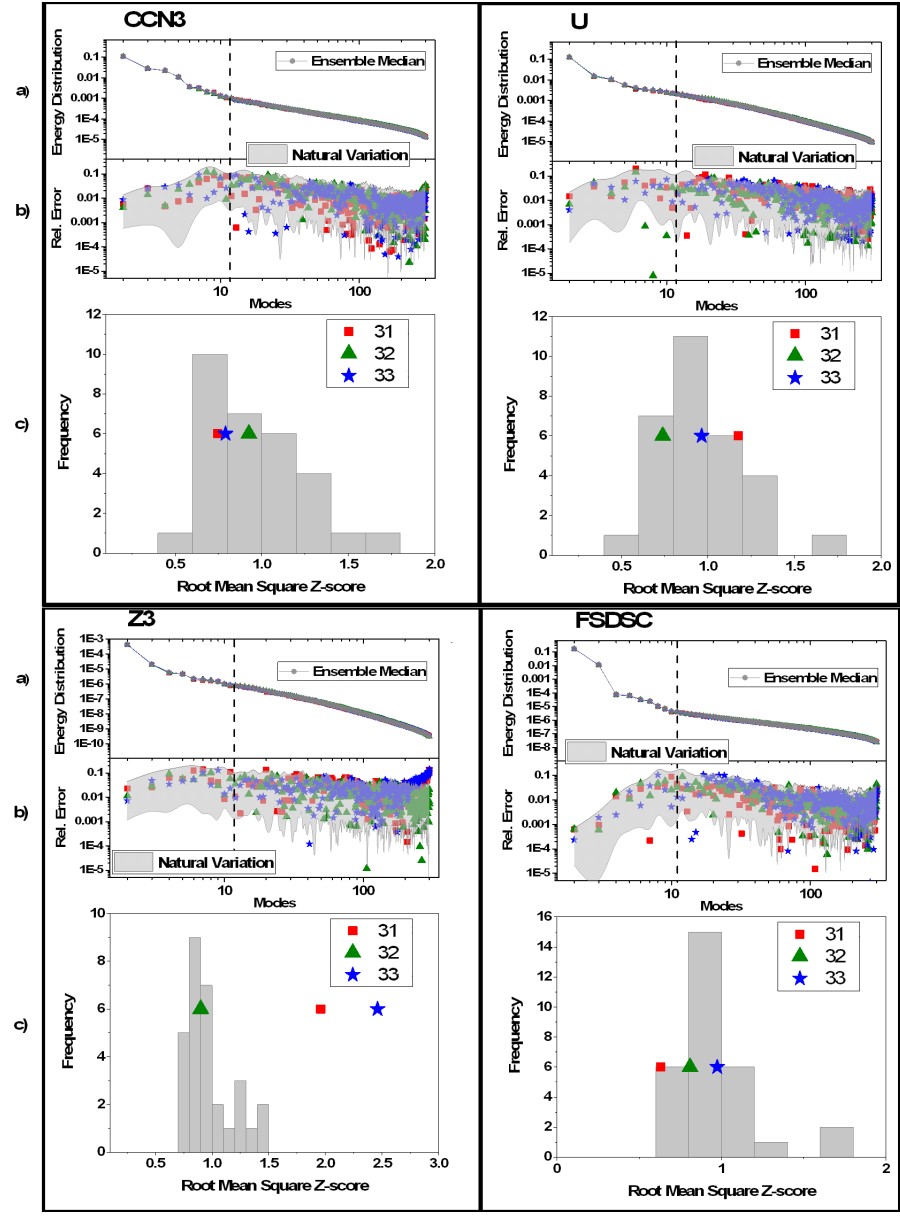

**Figure 16.** For each of the four variables studied, we show the following: a) Energy distribution of the modes of the POD for ensemble members 31-33, superimposed on the median of the original ensemble members (1-30); (b) relative errors of the energy distributions of the modes of the POD for new ensemble members 31-33 and the median of the original ensemble together with the natural variability observed within the uncompressed ensemble; (c) RMSZ distribution of the energy distribution for the 30 members of the original ensemble together with the RMSZ score of the energy distribution of new members 31-33.





**Figure 17.** GEV shape parameter $\xi$ variability, see Eq. (5.1.1). The left, middle and right panels correspond to the pdf of $\xi$, its range among compressed runs and its difference between a compressed and uncompressed run, respectively.





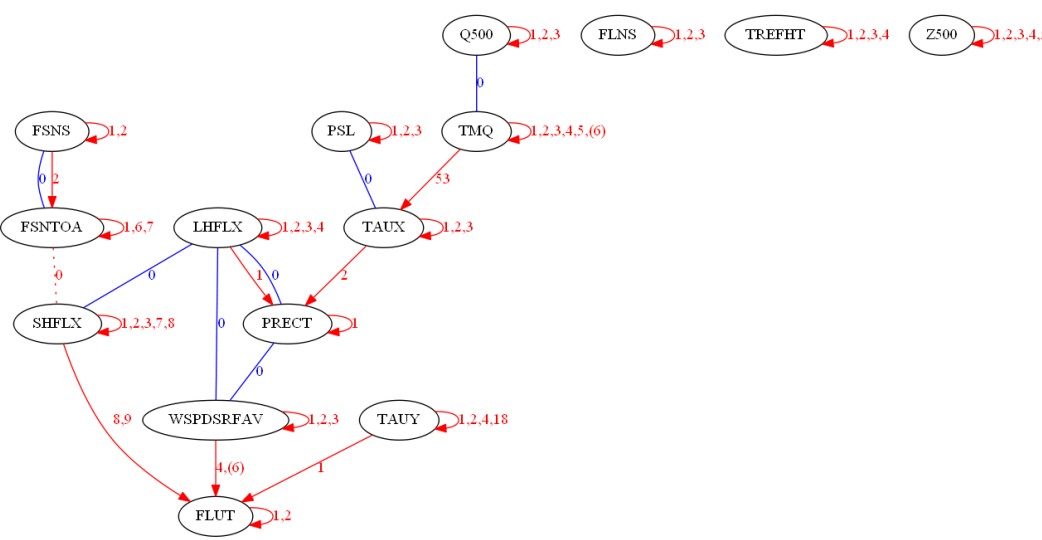

**Figure 18.** Causal signature interaction map for CESM-LE member 31. Blue lines delineate instantaneous connections and red lines indicate connections with a time lag. The number(s) next to each line give the number of days from potential cause to potential effect. The single dotted line between SHFLX and FSNTOA indicates a very weak instantaneous connection. Note that the causal signature for reconstructed CESM-LE member 31C is identical to this figure, except that the weak connection between SHFLX and FSNTOA is no longer present.

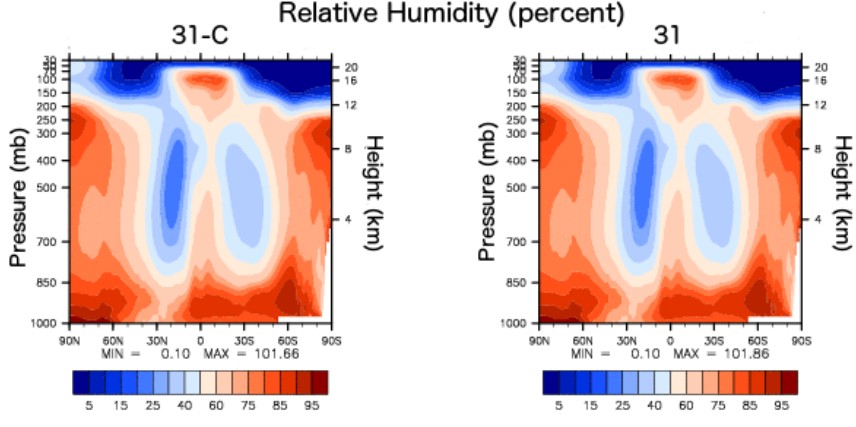

**Figure 19.** Vertical contour plot of DJF (December-January-February) zonal means for relative humidity (RELHUM) for 2006-2099 for ensemble member 31. The data in the left subplot has undergone lossy compression (i.e., 31-C) and the right subplot contains the original data.





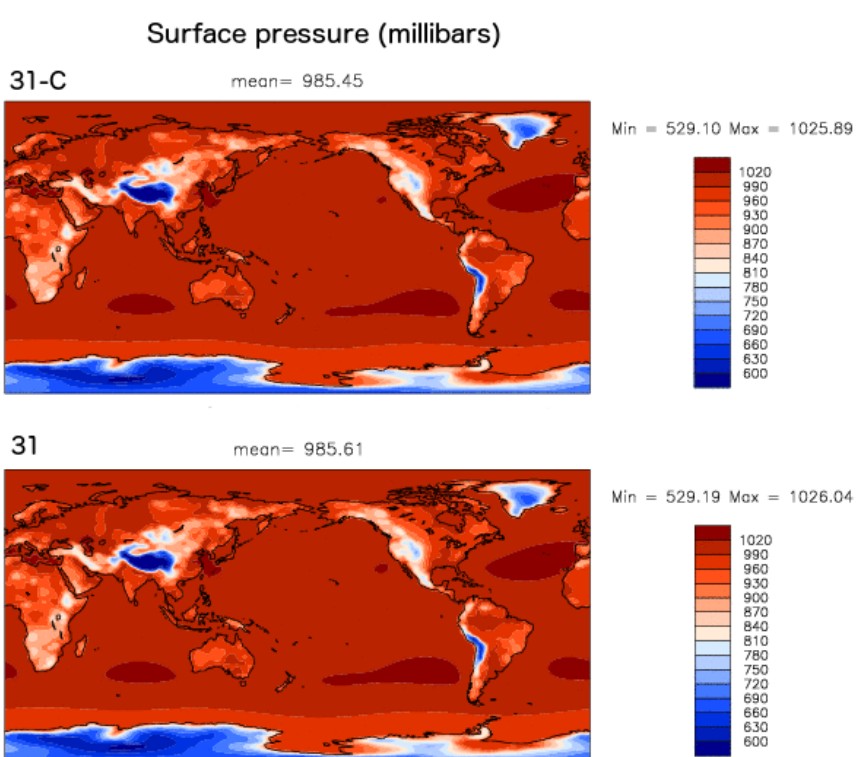

**Figure 20.** Horizontal contour plot of DJF (December-January-February) means for surface pressure (PS) for 2006-2099 for ensemble member 31. The data in the top subplot has undergone lossy compression (i.e., 31-C) and the bottom subplot contains the original data.

**Table 3.** Subset of AMWG Diagnostics Set 1: Annual Means Global. RESTOM and RESSURF are AMWG-DP derived variables for the top of model residual energy balance and the surface residual energy balance, respectively. RMSE indicates the root mean squared error. Units are $W/m^2$.

| Variable | Compressed Case | Original Case | Difference | RMSE |
|---|---|---|---|---|
| RESTOM | 2.016 | 2.016 | 0.000 | 0.001 |
| RESSURF | 1.984 | 1.984 | 0.000 | 0.000 |



**Table 4.** A list of co-authors and their corresponding evaluations.

| Section | Type of evaluation | Author(s) |
| --- | --- | --- |
| 4.1 | CVDP | Mickelson, Kay |
| 4.2 | Climate characteristics | Stolpe |
| 4.3 | Ensemble variability patterns | Sanderson |
| 4.4 | Coherent structures | De Simone, Carbone, Gencarelli |
| 5.1 | Climate extremes | Naveau |
| 5.2 | Causal signatures | Ebert-Uphoff, Samarasinghe |
| 5.3 | AMWG diagnostics package | Xu |