# Peer review of "Evaluating Lossy Data Compression on Climate Simulation Data within a Large Ensemble"

_Geoscientific Model Development, 2016_

## Referee Comment (RC1) · Anonymous Referee #1 · 12 Sep 2016

The topic of the paper is of high importance for data intensive computational sciences, in particular here for climate research. It discusses the possibility to use lossy compression for data volume reduction without harming the quality of scientific validity.

Lossy compression achieves higher compression rates than lossless compression and thus allows storage costs (investment and operational) to be reduced. For a single compute center that concentrates on weather and climate computations these savings will be a two-digit million USD amount with every new computer generation.

The paper written by a group of authers investigates the question to what extent lossyly compressed data can be detected in the subsequent steps of a scientific workflow. It does not discuss the question whether scientific results will be invalid.

The main idea is to introduce new data in ensemble computations where two new

ensemble members went through lossy compression and reconstruction. The study looks into whether these two ensemble members can be identified.

The authors group investigates 7 ways of how to potentially detect the new members that went through lossy compression. All 7 ways are different and sophisticated, thus a single reviewer will not be able to check the validity of each of the seven analyses.

In summary the authors present 4 lessons learned from the analyses. So, lossy compression seems to be possible and beneficial when certain conditions are met and the approach follows certain rules.

The paper presents a new research strategy for a highly important and exciting question. So there are several questions to the authors from a theory of science point of view and from a more meta science perspective.

1. When reading the paper it was not clear that the individual analyses are conducted by different authors who are specialists in their fields. This should be highlighted. Did they know of each other? Where they confronted with the research question independently? Please explain a bit the methodology of how the cooperation went.

2. Why exactly these 7 analyses? Do they somehow cover in a representative way what is done with the data or can be done with the data? Are there further analyses that should be added? Perhaps add some text at the end of Chapter 3 and describe the methodology of your approach (also for question 1).

3. There is not really a related works section with respect to this particular research. I assume in fact that there is not much related work. So: who of the community is looking into lossy compression and its effects on scientific validity of results? At least the GRIB people might do that? Any comparable efforts at other centers like NOOA, ECMWF, MetOffice, etc.? Please report. If there is no related effort, please confirm that in your paper.

4. What about the state-of-the-art with respect to algorithms? Google queries with

"lossy compression of climate data" and "lossy compression of medical image data" show that others also conduct research here. Lossy compression might be an issue in several other areas. Please give some details.

5. The analyses are a mixture of mathematical and visual approach: at first you apply a mathematical operation onto the data, then visualise it, then say that the effects of compression are discernable or not. Sometimes you mean: with the naked eye when looking at the charts? Is this a methodologically correct approach? Discernable depends on the person who looks at it.

6. fpzip seems to be old? Are there other algorithms with different approaches. Just like as for audio files there is quite some progress with lossy compression.

I completely agree with your conclusion at the end of page 4. Unfortunately, this is not clear to all researchers! You need to repeat it whenever there is a chance for it.

Now as we see, that lossy compression should be possible but is technically complicated because you would e.g. have to inspect all variables and decide upon what to do with them the questions are:

7. Should we proceed with looking into lossy compression as the advantage over lossless might only be a factor of 3 and with lossless there is no further problem?

8. What will be the extra costs in order to support lossy compression correctly with respect to human resources for e.g. variables analysis? Also there might be costs for additional hardware to do the compression efficiently and for power to operate this hardware. Of course, the benefits will be easier to quantify. However, please make a comment on the potential cost-benefit-ratio of introducing lossy compression into the science workflow.

A technical question to table 2:

9. Who/what defined the level that was used for each variable. Based on what consideration? Was this explained?

---

## Referee Comment (RC2) · Anonymous Referee #2 · 19 Sep 2016

General comments

The paper analyses in detail how lossy data compression affect climate simulation data. After choosing meticulously the level of reduction, they added 3 members in the CESM Large Ensemble project (2 with data loss compression and 1 without) and proposed to scientists to use there own tools to analyses these additional members. 4 different evaluations are summarized: CVDP climate variability diagnostic package, climate characteristics, ensemble variability patterns and coherent structures. They also proposed to compare in detail compressed with original data. 3 different analyses are summarized: climate extremes, causal signatures and AMWG-DP atmosphere working group diagnostics package. During the ensemble data evaluation, reduction factor for one variable has been detected and needed to be changed (from fpzip-16 to fpzip-24). Choices for reduction factor still an important step and required meticulous

action. During the original and reconstructed data analyses, nothing special appears regarding these 3 additional members. This very good news should help to increase the confidence of climate community for lossly and adequate compression. Each analyse is very well described and very clear. The choice of the reduction factor still critical and has to be analysed in detail for a new kind of simulation like the grid resolution, output frequency and numerical precisions as mentioned by authors. After these very complete analyses, the lossy data compression appears as a very useful step to reduce drastically the amount of data produced during climate simulation, especially for large ensemble simulation. It should be considered by all modelling group involved in CMIP6 or, at least, later in CMIP7.

This article is very useful and analyses described should help to convince climate community to use data lossy and adequate compression in there analyses to reduce storage required by large simulations like CMIP.

Specific comments

Fpzip tool for compression could be replaced by other smart tools. Lessons learned ie relationship between variables, detectable versus consequential, individual treatment of variables are very important and must be shared through CMIP modelling groups through the WIP: WGCM (Working Group on Climate Models) Infrastructure Panel.

Technical corrections

Paragraph 4.2.1 : a figure with the differences between the maximum and minimum monthly averaged temperature is probably missing.

Figures 2 3 4 5 6 7 8 9 10 11 and 13 should include 31-c and 33-c in legend instead of 31 and 33 to help understanding.

Figures titles should be completed:

• Figure 7 : add a reference to fpzip-24

• Figure 9 : add a reference to fpzip-16

• Figure 17 : add a reference to the 4 different variables showed

[Figure]

---

## Author Comment (AC1) · 17 Oct 2016

Thank you for your thorough review. We address all comments below.

———————— General comments ————————

We appreciate the thoughtful comments on the usefulness of this work.

———————— Specific comments ————————

(1) Regarding fpzip:

Our intent is to eventually develop a smart compression framework that any compression utility (with an appropriate API) could plug into. It may well be that different compression methods will be used on different subsets of CESM variables.

[Figure]

(2) Regarding sharing lessons learned with WGCM Infrastructure Panel:

Thank you for this suggestion. Our hope is that lossy compression will eventually be accepted for CMIP datasets.

——————— Technical corrections ———————

(1) Regarding paragraph 4.2.1 ("a figure with the differences between the maximum and minimum monthly averaged temperature is probably missing"):

Only Figure 2 accompanies this section. We intentionally did not include a second figure for Section 4.2.1, as we felt that the two sentences at the end of 4.2.1 sufficiently conveyed the important highlights. Because this paper has a large number of figures, we were careful to only include the figures that were the most useful (in our opinon).

(2) Regarding "Figures 2 3 4 5 6 7 8 9 10 11 and 13 should include 31-c and 33-c in legend instead of 31 and 33 to help understanding."

The convention in the manuscript is to label with the "-C" only in figures that were generated in a "not-blind" analysis. In the revision, rather than modifying the figures themselves, we instead added the text "Note that members 31 and 33 have been subjected to lossy compression." to the figure captions (as was already done for Figure 14) for clarification. In addition to the figures noted by the reviewer, we also added this clarifying text to Figure 16.

(3) Regarding Figure 7 ("add a reference to fpzip-24"):

We have modified the caption in Figure 7 to refer to the adjusted compression level as fpzip-24. We similarly adjusted the caption in Figure 10 for consistency.

(4) Regarding Figure 9 ("add a reference to fpzip-16"):

We have modified the caption in Figure 9 to note that fpzip-16 was used.

(5) Regarding Figure 17 ("add a reference to the 4 different variables showed")

We modified the caption in Figure 17 to list variables TSMN, TSMX, PRECT, and PRECTMX and give their meanings.

---

## Author Comment (AC2) · 17 Oct 2016

Thank you for your thorough review and suggestions for improvement. We address all comments and questions below.

(1) Regarding the individual analyses and methodology:

The individual analyses were indeed conducted independently by specialists in their fields. Participants were recruited in a number of ways: an open call for participation on the CESM Large Ensemble (LE) project web page; a verbal request at both the CESM-LE AGU session in 2014 and the CESM summer workshop in 2015 (and accompanying advertisements on posters at each); and direct e-mail to many scientists working with CESM data. We also approached certain individuals directly as we knew that they had expertise in complementary areas of interest that were not yet covered. All participants

were informed that multiple scientists were participating in the study, but all approached the question independently. We did not specify how the data should be analyzed, but asked participants to detail which ensemble members they believed to have been compressed and why. Recall that several of the analyses were not-blind, a decision that was made if we thought the particular analysis technique would provide more insight if given both the original and reconstructed data (e.g., the AMWG diagnostics package). Participants were made aware of other scientists' analyses after all feedback was received via an initial paper draft that was put together by the two lead authors.

In the revision, we updated Section 3 ("Approach") of the manuscript to include a better description of how the analyses were chosen and conducted.

(2) Regarding why these 7 analyses:

We chose these analyses for a variety of reasons, with the primary intent to give a sample of what types of post-processing analysis occur. For example, we targeted some participants based on their knowledge base (e.g., Phillipe Naveau for his expertise in extremes as we were aware of the concern over whether lossy compression would affect extremes). We also felt it necessary to include CVDP and AMWG analyses as these tools enjoy widespread use in the climate community as a first exposure to a data set. Other analyses were included simply because they were quite thorough and interesting to us. Categorizing the analyses we chose as being overall representative of what could be (or is) done with CESM as suggested is too strong, as there are too many possibilities for post-analysis of CESM simulation data. We feel that the analyses presented in the paper do give the reader a flavor for what is done and the concerns that different scientists may have when using a data set that has undergone lossy compression. The analyses also help illustrate our take-away messages in Section 6.

In the revision, we added more details about our selection process for the analysis in Section 3 (to also address reviewer comment (1)).

(3) Regarding the related works section:

In Section 2.1, we cite works that are investigating applying lossy compression to scientific data. However, as far as work particular to lossy compression of climate data, we are only aware of a few that we list below. The first two papers listed were discussed in an earlier paper by the first author (Baker et al. 2014) and certainly should have been cited again in this manuscript. As for the third, fourth, and fifth papers, we only became aware of these very recent papers after our initial submission of this manuscript. As far as a reference on the effects of lossy compression on the scientific validity of results, we are not aware of related work or comparable efforts in this area.

In the revision, we expanded the discussion in 2.1 to include the five references below. We also noted that we are unaware of any other studies that evaluate the effects of lossy compression on the scientific validity of climate simulation results.

J. Woodring, S. M. Mniszewski, C. M. Brislawn, D. E. DeMarle, and J. P. Ahrens. "Revisting wavelet compression for large-scale climate data using JPEG2000 and ensuring data precision." In D. Rogers and C. T. Silva, editors, IEEE Symposium on Large Data Analysis and Visualization (LDAV), pp. 31 - 38, 2011.

N. Hubbe, A. Wegener, J. M. Kunkel, Y. Ling, and T. Ludwig. "Evaluating lossy compression on climate data". In Proceedings of the International Supercomputing Conference (ISC '13), pp. 343-356, 2013.

M. Kuhn, Kunkel, J., and T. Ludwig."Data Compression for Climate Data." Supercomputing Frontiers and Innovations, 3 (1), pp. 75-94, June 2016.

J.D. Silver, and C.S. Zender. "Finding the Goldilocks zone: Compression-error trade-off for large gridded datasets." Geoscientific Model Development Discussions, pp. 1-13, July 2016.

C.S. Zender. " Bit Grooming: statistically accurate precision-preserving quantization with compression, evaluated in the netCDF Operators (NCO, v4.4.8+)" Geoscientific Model Development, pp. 3199-3211, September 2016.

In addition, for reference, we also added the following citations for recent lossless climate data compression work to Section 3:

X. Huang, X., Ni, Y., Chen, D., Liu, S., Fu, H. and G. Yang. "Czip: A Fast Lossless Compression Algorithm for Climate Data. International Journal of Parallel Programming." pp.1-20, 2016.

S. Liu, Huang, X., Ni, Y., Fu, H. and G. Yang, 2014. "A high performance compression method for climate data". In 2014 IEEE International Symposium on Parallel and Distributed Processing with Applications, pp. 68-77, 2014.

(4) Regarding a discussion of other lossy compression algorithms:

We did not explore using other compression algorithms in this work, as evaluating multiple state-of-the-art algorithms (and developing a methodology for such evaluations) on CESM data was the focus of the earlier work in Baker et al. 2014 entitled "A Methodology for Evaluating the Impact of Data Compression on Climate Simulation Data", which is referenced several times in this manuscript. The scope in this work is to provide a better understanding how the loss of information due to lossy compression affects the climate data from the perspective of post-processing analysis by scientists using the data (as opposed to simpler metrics common to the data compression community, e.g., root mean-squared error, peak signal-to-noise ratio, ...), and we don't believe that adding more text discussing the pros and cons of available lossy methods falls within our scope.

(5) Regarding the mixture of mathematical and visual approaches:

We agree that the analyses described are a mixture of mathematical and visual approaches. This mixture of techniques presented reflects the post-processing analysis that climate scientists perform in practice. Analysis by climate scientists often does involve an interpretation of visualized data (enabled by the CVDP or AMWG diagnostics tools, for example) that could be categorized as subjective for its dependence on the
person who looks at it. In this work, our approach was to allow the participants to analyze the data relevant to their interests in the manner of their choosing. We intentionally did not dictate a methodology (as noted in Section 1).

(6) Regarding the choice of fpzip:

An earlier paper by the first author (Baker et al. 2014) found fpzip to perform the best on climate data (as noted in Section 3). We note that we have evaluated many lossy methods, both mentioned in that 2014 manuscript and since, and we have not found any that perform as well on the climate data as a whole as fpzip.

(7) Regarding: "Should we proceed with looking into lossy compression as the advantage over lossless might only be a factor of 3 and with lossless there is no further problem?":

As acknowledged by the reviewer in the statement of question (6), most climate simulations already involve loss, whether in time via the chosen output frequency (e.g., daily, monthly, etc.), in space via the chosen resolution, or when writing output to disk (converting from double to float). For that reason, the reviewer's statement "with lossless there is no further problem" is perhaps more accurately stated as "with lossless we accept the loss in precision and resolution that has already been introduced into the process". In this light, we feel that it certainly makes sense to proceed with investigating the validity of the results after applying lossy compression. In fact, because the least significant bits in the simulation do contain error, this loss may even be desirable. For example, if a factor of 3 reduction due to lossy compression could be achieved with no impact on accuracy, then decimation in time and space could be less severe, and such a tradeoff could improve rather than degrade accuracy by discarding wasteful precision in favor of higher temporal or spatial resolution. Further, we note that achieving a factor of 3 (or even 2) reduction in data volume would be welcome news to data centers such as that at NCAR that are struggling with the financial burden of large climate simulation data volumes.

(8) Regarding the extra costs in order to support lossy compression:

While the focus of this work is determining whether lossy compression negatively impacts climate science results, we recognize that potential impact on the science workflow is of interest to many. We note that the energy cost of computation is negligible compared to the cost of data movement, e.g., in an arithmetic operation, 99% of the energy is spent moving the operands to registers from memory, while 1% is spent on performing the actual computation (e.g., Kestor et al., IISWC 2013). Therefore, we expect the energy cost of compression, even if done in software, to be insignificant compared to the energy cost of writing the data uncompressed to disk, and that using compression will in fact result in a net reduction in energy usage. As detailed in (Lindstrom and Isenburg 2006) and (Lindstrom et al. 2016, doi:10.1016/j.cageo.2016.04.009), I/O time is also substantially reduced by using compression. In practice, we note that the output data from the CESM-LE project is stored in compressed NetCDF format (lossless), which to our knowledge has not negatively affected user workflows. Ideally several lossy compression techniques will be incorporated into NetCDF in the future as well (we have had such discussions with Unidata).

(9) Regarding the level of compression for each variable (Table 2):

We describe how an appropriate level of compression was chosen in Section 3, beginning on page 6 (line 16) and continuing through page 7 (line 8). Please let us know if additional details are needed beyond the provided explanation.